# Families with BAP1-Tumor Predisposition Syndrome in The Netherlands: Path to Identification and a Proposal for Genetic Screening Guidelines

**DOI:** 10.3390/cancers11081114

**Published:** 2019-08-04

**Authors:** Cindy Chau, Remco van Doorn, Natasha M. van Poppelen, Nienke van der Stoep, Arjen R. Mensenkamp, Rolf H. Sijmons, Barbara W. van Paassen, Ans M. W. van den Ouweland, Nicole C. Naus, Annemieke H. van der Hout, Thomas P. Potjer, Fonnet E. Bleeker, Marijke R. Wevers, Liselotte P. van Hest, Marjolijn C. J. Jongmans, Marina Marinkovic, Jaco C. Bleeker, Martine J. Jager, Gregorius P. M. Luyten, Maartje Nielsen

**Affiliations:** 1Department of Ophthalmology, Leiden University Medical Center, 2333 ZA Leiden, The Netherlands; 2Department of Dermatology, Leiden University Medical Center, 2333 ZA Leiden, The Netherlands; 3Department of Clinical Genetics, Erasmus Medical Center, 3015 GD Rotterdam, The Netherlands; 4Department of Ophthalmology, Erasmus Medical Center, 3015 GD Rotterdam, The Netherlands; 5Department of Clinical Genetics, Leiden University Medical Center, 2333 ZA Leiden, The Netherlands; 6Department of Clinical Genetics, Radboud University Medical Center, 6525 GA Nijmegen, The Netherlands; 7Department of Genetics, University Medical Center Groningen, 9713 GZ Groningen, The Netherlands; 8Department of Clinical Genetics, Netherlands Cancer Institute, 1066 CX Amsterdam, The Netherlands; 9Department of Clinical Genetics, Amsterdam University Medical Centers, 1081 HV Amsterdam, The Netherlands; 10Department of Clinical Genetics, University Medical Center Utrecht, 3584 CX Utrecht, The Netherlands

**Keywords:** *BAP1*, BAP1 tumor predisposition syndrome, germline, referral guidelines

## Abstract

Germline pathogenic variants in the BRCA1-associated protein-1 (*BAP1*) gene cause the BAP1-tumor predisposition syndrome (BAP1-TPDS, OMIM 614327). BAP1-TPDS is associated with an increased risk of developing uveal melanoma (UM), cutaneous melanoma (CM), malignant mesothelioma (MMe), renal cell carcinoma (RCC), meningioma, cholangiocarcinoma, multiple non-melanoma skin cancers, and *BAP1*-inactivated nevi. Because of this increased risk, it is important to identify patients with BAP1-TPDS. The associated tumors are treated by different medical disciplines, emphasizing the need for generally applicable guidelines for initiating genetic analysis. In this study, we describe the path to identification of BAP1-TPDS in 21 probands found in the Netherlands and the family history at the time of presentation. We report two cases of de novo *BAP1* germline mutations (2/21, 9.5%). Findings of this study combined with previously published literature, led to a proposal of guidelines for genetic referral. We recommend genetic analysis in patients with ≥2 BAP1-TPDS-associated tumors in their medical history and/or family history. We also propose to test germline *BAP1* in patients diagnosed with UM <40 years, CM <18 years, MMe <50 years, or RCC <46 years. Furthermore, other candidate susceptibility genes for tumor types associated with BAP1-TPDS are discussed, which can be included in gene panels when testing patients.

## 1. Introduction

Germline pathogenic variants in the BRCA1-associated protein-1 (*BAP1*) gene underlie the BAP1-tumor predisposition syndrome (BAP1-TPDS, OMIM 614327) [1]. At first, somatic *BAP1* mutations were identified in 84% of 31 analyzed metastasized uveal melanoma (UM) and these mutations were found to be associated with metastatic progression of UM [2]. In one of the patients in this study by Harbour et al., a germline aberration in *BAP1* was found. This led to the suggestion that germline alterations in *BAP1* could predispose to development of UM. The following year, a pathogenic germline variant of *BAP1* was found in a family with multiple cases of UM [3]. Between 2011 and 2013, germline *BAP1* variants were also reported in association with cutaneous melanoma (CM) [4,5,6,7], malignant mesothelioma (MMe) [5,6,7,8], and renal cell carcinoma (RCC) [9,10].

Since then, several malignancies were found to be associated with *BAP1* germline pathogenic variants, the syndrome has been termed BAP1-TPDS [3,8,11]. More recently, the list of associated malignancies was expanded to include non-melanoma skin cancer (NMSC), meningioma, and cholangiocarcinoma [3,12,13,14,15]. NMSC encompasses basal cell carcinomas and cutaneous squamous cell carcinomas. Aside from malignant tumors, *BAP1*-inactivated nevi (BIN), previously called melanocytic *BAP1*-mutated atypical intradermal tumors (MBAITs) or BAP-oma [5,16], commonly occur in patients with BAP1-TPDS [17]. These are proliferations of atypical melanocytes deficient in BAP1 protein expression, often with spitzoid morphology [18]. Several other malignancies have been brought in association with BAP1-TPDS, for example: leptomeningeal melanoma [19], paraganglioma [20], and breast carcinoma [11]. However, a correlation was not confirmed in the largest studied cohort of 181 families with BAP1-TPDS to date [12]. The penetrance of pathogenic germline *BAP1* variants is high, with 85% of BAP1-TPDS individuals being diagnosed with ≥1 tumors [12,21].

Diagnostic testing of *BAP1* is needed to notify individuals carrying a pathogenic variant of *BAP1* and their families of their increased cancer risk. The occurrence of loss of function variants of *BAP1* in the general population is extremely low, with only 12 variants noted in gnomAD and the absence of homozygotes [22]. However, frequency of this dominantly inherited cancer syndrome has been reported as 1–2% of UMs [23,24,25], 0.5% of CMs [26], and 0–7% of MMes in unselected cases [8,27,28,29,30,31]. The detection of pathogenic variants in *BAP1* increases to up to 25%, 0.7%, and 20% when examining familial UM [24], CM [32], and MMe [33], respectively. The low frequency of pathogenic *BAP1* variants in unselected patients with BAP1-TPDS-associated malignancies implies that guidelines for the selection of patients with an increased risk of BAP1-TPDS are needed.

In this cohort study of BAP1-TPDS families in the Netherlands, we describe the various routes to identification of patients with germline *BAP1* pathogenic variants, their clinical phenotype, and genotype. Based on this information and review of the literature, we propose guidelines for when to perform genetic testing in individuals with an increased risk of having BAP1-TPDS. Moreover, a list of candidate susceptibility genes associated with BAP1-TPDS-related malignancies is provided that can be tested in addition to *BAP1*.

## 2. Results

### 2.1. Identification of BAP1-TPDS Families

In this cohort study, we describe 22 families with germline *BAP1* variants, that have been identified in the Netherlands. No clear guidelines for testing were available, but the combination of age at onset, spectrum of tumors and family history led to the suspicion of BAP1-TPDS. Previously, families NL-1, NL-3, NL-7, NL-12, NL-14, and NL-21 have been included in the cohort of Walpole et al. [12], but no extensive description of the families and mode of detection was provided. The original motivation for genetic analysis can be found in Table 1. In most cases, the analysis of the germline *BAP1* status was initiated as a result of a dermatological diagnosis. Eight of the 22 families (36%) were identified because of the observation of BINs, which were either the single tumor type (in five individuals) or were observed together with NMSC in the proband. Familial CM was the cause of genetic analysis in four families. Three families (NL-14, NL-16, and NL-20) (14%) were discovered in a research setting, where patients with familial CM were being studied. These families have been described by Potjer et al. [32]. Adding *BAP1* to the genetic analysis panel for familial CM in the Netherlands was recommended by the authors.

The combination of cutaneous and ocular melanoma resulted in the identification of four families (18%). The proband of family NL-8 was diagnosed with a conjunctival melanoma and five years later, with CM. This is the first report where conjunctival melanoma has been identified in a patient with BAP1-TPDS.

Four cases of UM in three consecutive generations triggered the ophthalmologist to refer the proband to the clinical geneticist. Furthermore, the occurrence of UM and ≥1 non-melanoma BAP1-TPDS-associated tumors led to the suspicion of an underlying genetic disease in four families (18%). The last variant was discovered in a proband with peritoneal MMe and a family history of MMe.

Thus, genetic analysis was initiated as a result of the medical history of the proband in 12 cases (55%) and the combination of medical history and family history of the proband in 10 cases (45%). Seventeen families were referred for genetic analysis based on a dermatological diagnosis treated by a dermatologist (*n* = 11), general surgeon (*n* = 1), a general practitioner (*n* = 1) and three families were identified through research of familial CM [32]. Two cases were referred by an ophthalmologist, one by an oncologist, and one by an internal medicine specialist. The family with variant c.437+1G>T (NL-18) was initially referred by a surgeon based on familial hepatocellular carcinoma, and the occurrence of UM and MMe in this family led to genetic analysis of *BAP1*.

### 2.2. BAP1 Variants in The Netherlands

In the 22 identified separate families with a germline *BAP1* variant in the Netherlands, 21 unique variants in *BAP1* were observed, including 11 variants that have not been reported previously. Thirteen germline *BAP1* variants resulting in a truncated protein were found in 14 families (Table 2). It is likely that the Dutch families, NL-10 and NL-11, with the identical *BAP1* germline variant have a common ancestor, however, no overlap in the family trees has been found.

Furthermore, six splice site variants and two missense variants in *BAP1* were found (Table 3). Splice site variant c.122+5G>C, p.? (NL-17) was classified as a variant of unknown significance (VUS), which led to additional RNA analysis to examine the effect on RNA splicing [34,35]. The splicing of the transcript was found to be affected by the germline variant: the donor site of exon 3 was not recognized, which resulted in the absence of the transcript of exon 3 of the aberrant allele. The diagnostic laboratory classified this variant as likely pathogenic based on these findings. Details of the RNA analysis can be found as Appendix A.

Variant c.200A>G, p.Asp67Gly (NL-21) is located in the protein domain Peptidase C12, ubiquitin carboxyl-terminal hydrolase 1. In silico predictions classify this variant as a VUS. Functional analysis of the germline *BAP1* variant led to the classification of this variant to be likely pathogenic [39]. This DNA change has previously been reported by Cabaret et al. in a patient affected by multiple BINs and a family history of mesothelioma [40]. Our patient has a medical history of UM and CM, with a family history of RCC (Table 1). Interestingly, all the core malignancies of BAP1-TPDS are present when combining the phenotypes of these families.

A novel missense variant c.1387C>G, p.(Leu463Val) was found in family NL-22, that was affected by multiple cases of CM. The genetic variant was classified as a VUS by the in-silico prediction models. However, the variant’s pathogenicity is difficult to determine without additional functional testing. This family will be excluded from further family analyses to ensure that any drawn conclusions are applicable to BAP1-TPDS. 

Two cases of de novo mutations were found in the cohort of BAP1-TPDS families. The mutational changes c.1017_1048del (NL-7) and c.1819delA (NL-13) in the germline *BAP1* were not found in parents and a sibling of both probands. In addition to the germline mutation in *BAP1*, another germline pathogenic variant was found in the proband of NL-7. The pathogenic variant in *ENG* (c.1116-1117insT, p.(Lys373*)) was inherited through the paternal lineage. Biological parenthood has been genetically confirmed in family NL-13. The *BAP1* variants are, therefore, a result of a de novo mutation, either in the proband or one of the parents, since there is a small chance that one of the biological parents of these probands harbors a mosaic *BAP1* germline pathogenic variant.

The frequency of probable de novo mutations in our cohort of BAP1-TPDS families is 2/21 probands (9.5%). This is the first report of the prevalence of de novo mutations found in the *BAP1* gene within a set BAP1-TPDS population, as far as we are aware. We have previously reported family NL-7 as part of a large international cohort of BAP1-TPDS individuals [12].

### 2.3. Clinical Characteristics of BAP1-TPDS Families in The Netherlands

Information on clinical characteristics was available from 727 individuals from the remaining 21 families affected by a pathogenic or likely pathogenic variant in *BAP1*. In addition to the 21 probands, germline *BAP1* status was determined in 85 family members, of whom 41 (48%) carried a (likely) defective *BAP1* allele. This led to the identification of ten additional BAP1-TPDS individuals as obligate carriers. A cohort of 72 carriers of germline pathogenic *BAP1* variants was established.

There are slightly more female (*n* = 43, 60%) than male (*n* = 29, 40%) carriers. The median age of 18 probands at *BAP1* testing was 43 years (range: 14–68). In the remaining three index patients, the germline *BAP1* variant was found after they passed away. In this study, the follow-up of the family is completed at diagnosis of BAP1-TPDS in the proband. The malignancies found in the proband and the family history of BAP1-TPDS-associated malignancies, before genetic analysis was performed, are listed in Table 1. The occurrence of the BAP1-TPDS core malignancies is illustrated in Figure 1 for all probands (a) and for the BAP1-TPDS families (b). Malignancies found in probands, tested non-proband carriers and untested members are presented in Appendix A. While most probands only exhibited one type of BAP1-TPDS-associated malignancy, the phenotype found in the families are more diverse.

### 2.4. BAP1-TPDS Core Malignancies

#### 2.4.1. Uveal Melanoma

In our BAP1-TPDS population, nine of the 72 carriers of germline *BAP1* (likely) pathogenic variants (13%) were affected by UM. The median age at diagnosis in these patients was 61 years (range 30–72), which is similar to the median age of 66 years found in the general UM population in the Netherlands [41].

Most (*n* = 7, 78%) UMs were primarily located in the posterior segment, although one patient was diagnosed with iris melanoma. Six eyes with UM were enucleated, two tumors received ruthenium brachytherapy and one tumor was treated with stereotactic radiotherapy. Five BAP1-TPDS patients developed metastases from their primary UM (56%, median time to metastases from diagnosis: 24 months, range: 0.2–67 months), while four patients did not (median time since diagnosis: 25 months, range: 21–184 months). The occurrence of UM in combination with another BAP1-TPDS-associated malignancy was observed in three patients: other malignancies included both a RCC and a MMe in one patient and CM in two patients. These other malignancies were all diagnosed after diagnosis of UM.

Of the untested relatives, six individuals were reported to have UM. Furthermore, four cases of (unknown types) of ocular tumors have been listed in the family history by patients (Appendix A).

#### 2.4.2. Cutaneous Melanoma

CM was the most frequently diagnosed malignancy among the current cohort of BAP1-TPDS individuals in the Netherlands, with 15 tumors occurring in 13 of the 72 carriers (21%). The median age at diagnosis of CM in this group was 49 years of age (range 23–83). This is younger compared to the median age of onset of 63 years in the general CM population in the Netherlands [41].

The median Breslow thickness was 1.4 mm (range: 0.34–2.8 mm), calculated from the known sizes of 11 CMs. Metastatic disease of CM was found in three of the 13 CM patients.

Eighteen CMs were reported in untested family members. Interestingly, five CMs were found in three tested relatives without the familial *BAP1* germline aberration. These patients are, therefore, considered to be phenocopies.

#### 2.4.3. Malignant Mesothelioma

Eight BAP1-TPDS patients were found to have MMe (8/72 BAP1-*TPDS* individuals, 11%). The median age of onset was 60 years (range: 39–71), which is earlier than the median age of 74 years at diagnosis in the general Dutch population [41]. MMe was classified as pleural in six cases (75%) and peritoneal in two cases (25%).

In untested relatives, MMe was reported in nine cases: five pleural MMe and four peritoneal MMe. Notably, 14 of the 17 MMe patients were distributed over three families. Both family NL-4 and family NL-18 had three cases of MMe, while eight individuals with MMe were found in family NL-2. In the family NL-18, MMe was observed in two out of four siblings. In the family NL-2, six out of 12 siblings were diagnosed with MMe.

#### 2.4.4. Renal Cell Carcinoma

Two of the 72 proven BAP1-TPDS patients (3%) developed RCC. They were diagnosed at 58 and 61 years of age, which is lower than the median age of 68 in the general Dutch population [41]. Both patients also had MMe. One of the two had previously been diagnosed with UM.

#### 2.4.5. BAP1-Inactivated Nevus

We observed 20 histologically proven excised BINs in eight BAP1-TPDS probands. The median age of onset was 21 years of age (range: 14–55). Seven patients had two or more confirmed BINs before diagnosis of BAP1-TPDS, with a maximum of five BINs in one patient.

#### 2.4.6. Other Malignancies

New malignancies associated with BAP1-TPDS, as suggested by Walpole et al., include meningioma, cholangiocarcinoma, and non-melanoma skin cancer (NMSC) [12]. No case of cholangiocarcinoma was reported in our patient population and meningioma was present in one proven carrier of a pathogenic *BAP1* gene (1/72, 1.4%), who developed a vestibular schwannoma as well as an UM later in life. An untested family member of this patient was also diagnosed with both UM and meningioma.

The frequency of NMSC was substantially higher compared to meningioma and cholangiocarcinoma, with 19 affected BAP1-TPDS patients (19/72, 26%). Most NMSC were basal cell carcinomas, although the exact distribution of NMSC is unknown. The median age of onset was 50 years (range: 27–67). The number of tumors varied between 1 and 36, with a median of 2 NMSCs.

The described cohort in this study is too small to draw conclusions about possible associations of other tumors types to pathogenic *BAP1* variants. However, we did notice the occurrence of primary non-cirrhotic hepatocellular carcinoma in three proven *BAP1* germline variant carriers in family NL-18.

### 2.5. Disease-Free Survival

The disease-free survival of our included study population is plotted in Figure 2 for UM (a), CM (b), MMe (c), and RCC (d). An event is defined as a tumor occurrence. Follow-up starts at birth, since individuals with BAP1-TPDS are at risk of developing a malignancy from birth. The red line illustrates the disease-free survival of non-proband BAP1-TPDS individuals. The disease-free survival of all pathogenic *BAP1* variant carriers is shown for reference (gray dotted line). Follow-up data after the age of 65 years is unreliable, due to the limited number of non-proband germline *BAP1* variant carriers. Disease-free survival curves including and excluding the probands do not differ clearly for UM (*p* = 0.19), CM (*p* = 0.26), MMe (*p* = 0.95), and RCC (*p* = 0.75) at the age of 65 years, as can be seen in Figure 2.

The cumulative disease-free proportions and their corresponding 95% confidence interval (95% CI) were estimated at the age of 65 years, as with increasing age, estimated proportions were captured with more uncertainty. The cumulative disease-free proportion at age 65 was 0.93 (95% CI 0.79–1) for UM, 0.85 (95% CI 0.73–0.97) for CM, 0.74 (95% CI 0.54–0.95) for MMe, and 0.94 (95% CI 0.84–1) for RCC.

## 3. Discussion

In this study, we have described the diverse routes that led to the identification of germline *BAP1* pathogenic variants in probands and relatives of 22 BAP1-TPDS families in the Netherlands. We have found that medical history and family history led to the diagnosis of BAP1-TPDS in the described families. Genetic analysis was most often initiated as a consequence of BINs in the medical history of the proband. De novo germline *BAP1* variants were found in families NL-7 and NL-13 (2/21, 9.5%), in both cases, paternity was confirmed. The first presentation of these families occurred in various medical fields, which emphasizes the need for attentive physicians to recognize potential germline *BAP1* variant carriers.

### 3.1. BAP1-TPDS-Associated Malignancies

In the current study, we describe the phenotype of BAP1-TPDS families and individual *BAP1* germline pathogenic variant carriers in the Netherlands at the time of presentation to the clinical geneticist. No additional immunohistochemistry or genetic analysis of tumors in the carriers of germline *BAP1* variants has been performed. Consequently, it is uncertain if the development of the found malignancies are correlated to the aberrant *BAP1* allele. This is an interesting field for future research.

#### 3.1.1. Different Types of Melanomas

While UM and CM are genetically distinct malignancies, both have been established as BAP1-TPDS-associated malignancies. Although UMs include iris melanomas, this is the first observation of an iris melanoma in a patient with BAP1-TPDS. Scholz et al. observed somatic mutations in *GNAQ*, *GNA11*, *EIF1AX*, and *BAP1* (four genes that are often mutated in choroidal and ciliary body melanomas [42]) in 19 iris melanomas [43]. When a larger cohort of 30 iris melanomas was analyzed by Van Poppelen et al., these findings were confirmed, with additional mutations observed in *SF3B1,* but also in the CM-associated genes *NRAS*, *BRAF*, *PTEN*, *c-KIT*, and *TP53* [44,45,46]. Although the genetic basis of iris melanomas is still being studied, iris melanoma could be considered as a BAP1-TPDS-associated malignancy.

Conjunctival melanoma is a rare ocular malignancy occurring in the mucosal surface of the eye, with an incidence of 0.2–0.8 per million in Caucasians [47,48]. Although conjunctival melanomas are located on the mucosal surface of the eye, the melanomas located in the conjunctiva have been identified as being more genetically similar to CM than UM. Mutations in the CM driver genes *BRAF* and *NRAS* have been found in conjunctival melanoma [49,50]. The concurrence of conjunctival melanoma and BAP1-TPDS is remarkable in light of the low prevalence of both conditions. This is the first reported case of mucosal melanoma in association with BAP1-TPDS. However, a causal correlation is not proven due to the absence of molecular testing or BAP1 immunohistochemical staining. This matter falls outside the scope of this article, as tumor tissue analysis was not performed before germline *BAP1* analysis. However, it is a relevant topic to explore in future studies.

Conjunctival and other mucosal melanomas have not been included in the BAP1-TPDS spectrum and larger cohorts of BAP1-TPDS individuals need to be studied to determine a possible association. However, one may consider these tumors as potentially BAP1-TPDS-associated malignancies when deciding to refer a patient for genetic analysis. An alternative is to apply somatic DNA sequencing—or, if unavailable, *BAP1* immunohistochemistry—on tumor material to help decision making.

#### 3.1.2. Mesothelial Malignancy

In the general population, MMe is located in the pleura in approximately 82% of the cases. Previously, it has been noted that in the BAP1-TPDS population, the distribution of pleural MMe and peritoneal MMe is more equal [33,51]. In line with previous reports, we report a higher frequency of a peritoneal localization of MMe compared to the general population, with 11 pleural and 6 peritoneal MMes in genotyped and ungenotyped BAP1-TPDS family members.

The majority of MMes (14 out of 17 cases) in our study occurred within three families. In two of these families, half of a generation was affected by MMe: 2/4 siblings in NL-18 and 6/12 siblings in NL-2. This could imply that there is a genotype–phenotype relation between the specific *BAP1* aberration and the type of malignancy. However, this does not explain the increased observed frequency in one specific generation compared to the other generations. Furthermore, earlier reports in large BAP1-TPDS cohorts do not support this theory [12,21]. An alternative hypothesis is a gene–environment interaction.

Development of MMe is closely associated with asbestos exposure [52,53]. When individuals who are already more susceptible to develop MMe are exposed to asbestos, it could trigger the oncogenesis of MMe. While this mechanism has been suggested by researchers, it has not yet been proven in BAP1-TPDS. However, mouse models have been established where mice harbor a heterozygous pathogenic variant in *BAP1*. It has been observed that two of these mice out of a total of 93, spontaneously developed MMe [54]. MMe was found significantly more often in *BAP1*^+/−^ mice compared to the *BAP1*^WT^ littermates (73% vs. 32%, respectively), when exposed to asbestos fibers. Moreover, MMe arose in a shorter period of time in the *BAP1*^+/−^ mice after initial exposure (43 weeks vs. 55 weeks, respectively) [55]. While more research is needed in humans, this study suggests that while spontaneous development of MMe can occur, a gene–environment interaction could possibly explain the clustering of MMe in our BAP1-TPDS population.

#### 3.1.3. Renal Cancer

Even though only two cases of RCC were observed, both patients also exhibited MMe in their medical history. However, no conclusions can be drawn from this information in the described cohort.

#### 3.1.4. Non-Melanoma Skin Tumors

The most prevalent tumors associated with BAP1-TPDS in our population are NMSCs and BINs. BIN is a skin lesion that is closely associated with BAP1-TPDS, while the diagnosis might be missed outside of this context, as they can be indistinguishable from ordinary dermal nevi. These melanocytic nevi usually lack malignant characteristics clinically and can only be proven histopathologically. Busam et al. have reported the distinct histopathologic profile of BINs: these skin tumors are “characterized by a nevus-like silhouette and cytologic composition of large epithelioid melanocytes with oval vesicular nuclei, distinct nucleoli, and abundant cytoplasm with well-defined cytoplasmic borders” [56] (p. 193). Loss of nuclear BAP1 immunohistochemical staining is typical in BIN [56]. BINs are by definition benign. Malignant transformation has been described, but is rare [7,57,58]. Because of the predominantly benign prognosis, BIN-suspected lesions are often not removed.

Because of the high prevalence of both basal cell carcinomas and cutaneous squamous cell carcinomas in the general population and the unfamiliarity of the generally benign BINs, the diagnosis of BAP1-TPDS might frequently be missed. Education for dermatologists and pathologists concerning this dermatological diagnosis is necessary to establish the correct diagnosis.

#### 3.1.5. Tumor Frequencies Compared to International Cohorts

Nine cases of UM were found in 72 BAP1-TPDS patients (13%), 13 carriers with BAP1-TPDS were diagnosed with CM (18%), 8 with MMe (11%), and 2 with RCC (3%), while the recent worldwide meta-analysis found an occurrence of 25%, 20%, 20%, and 6% for UM, CM, MMe, and RCC, respectively [12].

A possible explanation of the lower numbers found in this study could be that our patient population is relatively young and not yet at an age to develop these malignancies. The mean age at *BAP1* germline analysis of the proband (or age at last follow-up in obligate carriers) was 44 years in the 42 carriers of a *BAP1* pathogenic variant without BAP1-TPDS core malignancies and 59 years in the remaining 27 BAP1-TPDS patients (age at *BAP1* analysis is missing for three patients) that have been diagnosed with UM, CM, MMe and/or RCC (*p* < 0.001). Another sign that points to age as the cause of the discrepancy is that the frequency of CM found in our population approaches the internationally reported frequency of CM in BAP1-TPDS. CM generally occurs at a younger age, compared to most other malignancies [59].

It is important to note that the frequencies of malignancies are reported based on a highly selected group of patients in both previously published articles and this study, where results are very likely to have been influenced by bias. While our patients have not been selected following a study protocol, it is likely that the referring physician noticed an unusually high frequency of BAP1-TPDS-associated malignancies in the probands and/or their family history. Testing bias plays a role as well, as patients with a previous medical history of BAP1-TPDS-associated malignancies are more likely to undergo genetic testing.

The clear disadvantage of using BAP1-TPDS probands for malignancy risk estimation is that these probands are mostly tested because of the presence of a malignancy. This will give an overestimation of malignancy risk when applied to an unselected BAP1-TPDS cohort. Therefore, we put most emphasis on non-proband *BAP1* variant carriers in the Kaplan–Meier curves. The highly selected probands from the BAP1-TPDS families are excluded to correct for selection bias in an attempt to present more representative frequencies of malignancies in BAP1-TPDS. However, the number of non-proband BAP1-TPDS individuals is limited, which results in wide 95% confidence intervals. In future research, a larger cohort is required to provide more precise estimates.

### 3.2. Guidelines for Referral for Genetic Analysis

As probands differ in their initial clinical presentation, it is necessary to provide guidelines that can be applied to different medical fields. Previously, Rai et al. have made recommendations for genetic assessment based on ≥2 BAP1-TPDS-associated tumors in the medical history and/or family history of an individual [21]. However, carriers of a germline *BAP1* pathogenic variant tend to develop BAP1-TPDS-associated malignancies at a younger age [12,21]. This observation, combined with the frequency of 2/21 (9.5%) of de novo mutations as reported in the current study, led to a proposal of age-associated criteria. The proposed referral guidelines are shown in Table 4.

We have adopted the recommendations made by Rai et al. concerning the medical history and family history of BAP1-TPDS malignancies [21], and we have added age-associated criteria. Since the publication of these recommendations, the spectrum of malignancies has expanded. One of the added malignancies is NMSC. As NMSC occurs very frequently in the general population, we classify non-melanoma skin cancers as a relative criterium, where referral can be considered when a high frequency of non-melanoma skin cancers is found in a single individual or at an unusually young age.

When CM is the only BAP1-TPDS-associated malignancy present in an individual or family, the threshold for familial CM should be ≥3 cutaneous melanomas in one individual or in the family, given its high frequency in the general population in the Netherlands. However, in populations where CM is less frequently found, it can be considered to uphold the criterium of ≥2 CMs. Genetic analysis for frequently affected CM susceptibility genes, like *CDKN2A* and *CDK4,* is indicated for these patients [32,63,64]. Testing of *BAP1* should be done in parallel with testing of these susceptibility genes, or performed subsequently in case no pathogenic variant is found.

In our study population, most of our probands (8/22, 36%) were identified as a consequence of BINs. Interestingly, Cabaret et al. found 12 probands harboring pathogenic germline variants in *BAP1* in a group of 17 (71%) patients with ≥1 BINs and a suspicious medical or family history [40]. In four families, BINs were the only phenotypical manifestation of BAP1-TPDS. Furthermore, Haugh et al. performed a review of the literature and discovered that BINs were found in 40/53 individuals with BAP1-TPDS that underwent total body inspection of the skin (75%) [65]. If a patient is diagnosed with a single BIN, without other malignancies in their personal medical history or family history, the criteria for germline testing are not met. However, BINs might be underdiagnosed as we have stated previously. Therefore, it is recommended to perform immunohistochemical staining of nuclear BAP1 or somatic analysis of *BAP1* in previously removed dermatological lesions to establish if germline analysis of *BAP1* is necessary.

The age-associated criteria are based on several previously published articles. Walpole et al. showed a median age at diagnosis for UM in patients with germline null variants of 53 years (interquartile range: 44–60 years) and a median age of 58 years for missense variants, (interquartile range: 45–69 years) [12], which are both under the reported median age of incidental UM of 66 years in the Netherlands [41]. Similarly, Rai et al., Haugh et al., Ewans et al., and Gupta et al. report younger ages in patients with BAP1-TPDS [21,23,65,66]. Although this difference was not significant when compared to control subjects by Gupta et al. (*p* = 0.44) and approaches significance compared to individuals with somatic *BAP1* mutations in UM (*p* = 0.07) [23,66]. Previously, the group from Abdel-Rahman et al. have proposed an age-associated limit of 30 years at diagnosis of UM [3,67,68]. However, as the BAP1-TPDS patients with UM in our cohort developed UM at the age of 30 years and older, we propose to extend this criterium to an age of onset of 40 years. 

Unfortunately, due to the lack of reported studies, suggestions for the limits of young age of onset for meningioma and cholangiocarcinoma cannot be produced. Future research is needed to provide age-associated criteria for these malignancies. In general, future cohort studies are needed to evaluate the efficacy of the proposed testing strategies, and adjustments in the guidelines should be made where needed.

When applying the guidelines to the studied cohort, 21 out of the 22 described families would have been identified. The family harboring VUS c.1387C>G, p.Leu463Val (NL-22) would not have been genetically analyzed based on the proposed guidelines. The largest described cohort of 181 families carrying germline *BAP1* variants has been published by Walpole et al. [12]. Unfortunately, the ages at diagnosis of malignancies and specification of family history (first-, second-, third-degree relatives) were not described in the article. If we apply the proposed guidelines based on BAP1-TPDS-associated malignancies to these families, assuming reported family members are first- or second-degree relatives, 120/141 (85%) families carrying null variants and 18/40 (45%) families carrying missense variants would have been offered genetic analysis. However, none of the families with missense variants that have been classified in the paper as having evidence towards pathogenicity would have been missed.

In the majority of families with multiple cases of one type of a BAP1-TPDS-associated malignancy, no pathogenic germline aberration was found in *BAP1*. The reported frequencies of a pathogenic germline *BAP1* variant in familial UM are 20% and 25% [24,69], in familial CM 0.7% [32], and in MMe 7.7% and 20% [33,70]. In 28 families with multiple members affected by RCC, but without pathogenic variants in the Von Hippel–Lindau gene, no pathogenic variants were found in *BAP1* [71]. Possible deep intronic variants in the *BAP1* gene, not detectable with standard diagnostic techniques, might be present in a small subset [25]. Other candidate susceptibility genes have been mentioned for most BAP1-TPDS-associated tumors. Some are well known and strongly associated, like *CDKN2A* for melanoma [72,73], while others are rare (*MBD4*) or more controversial, like *BRCA1/2* [74,75,76]. These genes require further research in larger cohorts. Meanwhile, several genes might be considered for inclusion in gene panels, as suggested in Table 5.

### 3.3. Proposed Surveillance Guidelines in BAP1-TPDS

The penetrance of *BAP1* germline pathogenic variants is reported to be 85% [12,21]. To date, all studies have been performed in a highly selected group of individuals, which may lead to an overestimation of the cancer risk. Until more information is available for BAP1-TPDS individuals, screening for BAP1-TPDS-associated malignancies is necessary. Surveillance in BAP1-TPDS patients could lead to early diagnosis and treatment of malignancies, which can potentially lead to prolonged survival. 

In 2014, Pilarski et al. suggested a multidisciplinary approach, recommending annual ophthalmological and dermatological evaluations, while clear guidelines for MMe and RCC were missing [11]. Two years later, more complete management recommendations were proposed for families with BAP1-TPDS by the same research group [21]. In addition to the examinations by an ophthalmologist and dermatologist, yearly physical examinations for MMe and a renal screening protocol adjusted from the renal screening for Von Hippel–Lindau syndrome was advised (annual abdominal ultrasound, MRI every two years). The reported ages of the youngest cases of BAP1-TPDS with BAP1-TPDS core malignancies were 16, 25, 34, and 36 years for UM, CM, MMe, and RCC, respectively. Initiation of the surveillance of the relevant malignancy was proposed to be 5 years before the first reported age at diagnosis of the malignancy in association with BAP1-TPDS.

Star et al. proposed a surveillance plan for BAP1-TPDS patients [86]. Similar to prior suggestions, they propose ophthalmological and dermatological evaluation. However, the authors advised to decrease the period between examinations to 6 months for the skin, starting at 18 years. Similarly, they advised to perform ocular examinations annually, starting at 16, expanding to twice a year, from the age of 30 onwards. At that age, annual physical examination was advised, to start to look for signs of MMe and/or RCC. Additionally, biennial radiological examination may be considered according to the Australian colleagues. The selection of the appropriate method of imaging is complex, as the evidence for screening in BAP1-TPDS has not been completed yet. The proposed surveillance recommendations by Rai et al. and Star at al. are listed in Table 6 [21,86].

In order to minimize the burden for patients, all annual screening should preferably be performed in one day, where patients are screened for BAP1-TPDS-associated malignancies by the ophthalmologist, dermatologist, and radiologist. An experienced multidisciplinary team is necessary to manage patients with this diverse spectrum of conditions.

## 4. Materials and Methods

At the Leiden University Medical Center, a multidisciplinary BAP1-TPDS outpatient clinic has been organized for screening of carriers with BAP1-TPDS. Patients were informed about the current study and gave consent for participation. This included written permission to review their medical record. A collaboration was established with several medical centers in the Netherlands who provided information from families with (likely) pathogenic germline *BAP1* variants found at their center. This study is a multicenter retrospective case series. 

### 4.1. Clinical Mutational Data

DNA diagnostics of *BAP1* were performed at the Leiden University Medical Center, Erasmus Medical Center, University Medical Center Groningen and Radboud University Medical Center. Data from germline *BAP1* variants were obtained from the medical records. Sequencing of *BAP1* was performed following the clinical protocol at each center. Reference sequence NM_004656.3 was used to determine the Human Genome Variation Society (HGVS) nomenclature on the DNA level. Chromosomal locations were determined using genomic build GRCh37/hg19. Interpretation of the reported variants was performed by the above-mentioned accredited clinical genetic centers of the Netherlands. This implies that VUS were only reported if there was a strong suspicion for possible pathogenicity in in silico predictions. 

### 4.2. Clinical Review

The following clinical data were collected from medical records: gender, results of genetic testing, medical history, age at time of diagnosis of tumors and at last follow-up, family history of tumors, and pathology reports. The medical information was collected by the treating physicians. To establish a correct representation of BAP1-TPDS families’ phenotype at the time of genetic analysis, only tumors that occurred prior to the diagnosis of BAP1-TPDS were included.

### 4.3. Survival Curves

Kaplan–Meier curves were plotted using IBM SPSS statistics 25.0, and compared with log-rank tests.

## 5. Conclusions

We described a cohort of BAP1-TPDS individuals from 22 families in the Netherlands and show a high cancer risk, in concordance with previous reports. The included families harbored 21 unique germline *BAP1* variants. A de novo *BAP1* mutation was present in 9.5% of the probands. We reported the first cases of iris melanoma and conjunctival melanoma in association with BAP1-TPDS.

The various routes to diagnosis of BAP1-TPDS have been illustrated and lead to a proposal for referral guidelines for germline *BAP1* analysis. Furthermore, alternative candidate susceptibility genes were summarized and can be used for gene panels for patients when genetic analysis is indicated based on young age at diagnosis or a single type of BAP1-TPDS-associated malignancy in their personal or medical history.

## Figures and Tables

**Figure 1 cancers-11-01114-f001:**
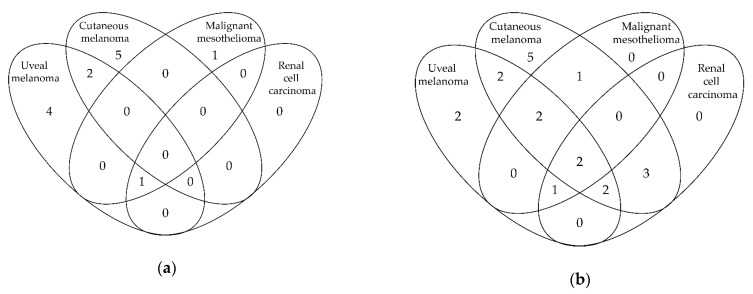
Venn diagrams of BAP1-tumor predisposition syndrome (BAP1-TPDS)-associated core malignancies found in BAP1-TPDS probands (**a**) and in the 21 BAP1-TPDS families (**b**). Eight probands were genetically analyzed as a result of the diagnosis of BINs and are, therefore, missing in Figure 1a.

**Figure 2 cancers-11-01114-f002:**
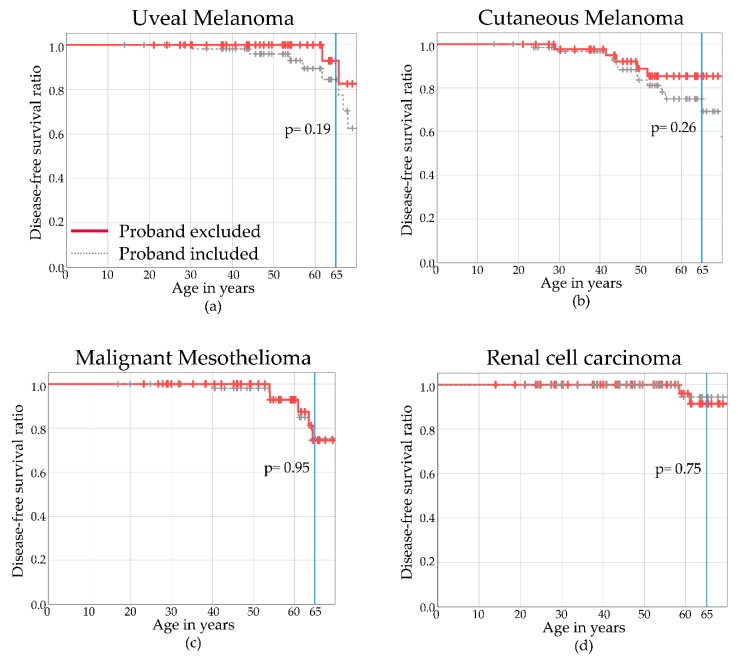
Disease-free Kaplan–Meier curves for proven/obligate non-proband *BAP1* variant carriers excluding probands (red) of uveal melanoma (**a**), cutaneous melanoma (**b**), malignant mesothelioma (**c**), and renal cell carcinoma (**d**). Individuals with BAP1-TPDS including probands are illustrated with the gray dotted line.

**Table 1 cancers-11-01114-t001:** The reason for genetic testing and the phenotype of BAP1-TPDS-associated malignancies in *BAP1*-variant carriers in The Netherlands.

Family Identifier	Reason for Testing	Tumors Proband (Age)	BAP1-TPDS-Associated Malignancies in Family History (Excl. Proband)
NL-1	UM and family history of multiple UMs	UM (67)	UM ×3, NMSC ×2
NL-2	Multiple BINs	BIN × (39 ×2) NMSC (39)	UM, CM ×4, MMe ×7, NMSC ×5
NL-3	Multiple BINs	BIN ×5 (29 ×5)	CM, NMSC ×4
NL-4	MMe and family history of MMe	MMe pt (39)	CM, MMe 2×, NMSC 2×
NL-5	UM & meningioma and family history of UM, CM, meningioma, and RCC	UM (66) NMSC (66) Meningioma (44) VS (51)	UM, CM, RCC, Meningioma
NL-6	UM and family history of MMe	UM (30)	CM ×2, MMe, RCC, NMSC ×9
NL-7	Multiple BINs	BIN ×2 (22 ×2)	CM ×2, NMSC ^b^
NL-8	CM & conjunctival melanoma	CM (49) CoM (44) Lung cancer (54)	CM, RCC, NMSC
NL-9	UM, MMe, and RCC	UM (57)—iris MMe pl (61) RCC (61) NMSC (57) B-cell lymphoma (58)	RCC
NL-10	Multiple BINs	BIN ×2 (20, 26) NMSC (27)	CM ^a^ ×2, NMSC ×4
NL-11	Single BIN	BIN (55) Breast cancer (48)	UM, CM, NMSC ×13
NL-12	UM & CM	UM (53) CM (56) NMSC (38, 52)	CM, MMe, RCC
NL-13	Multiple BINs	BIN ×4 (15 ×2, 18 ×2)	-
NL-14	Familial CM ^b^	CM ×2 (23, 27) NMSC ×2 (50, 53)	CM ×6, NMSC ×47, metastatic melanoma
NL-15	Multiple BINs	BIN ×2 (21, 25)	UM
NL-16	Familial CM ^b^	CM (65)	CM ×3, RCC
NL-17	Multiple BINs	BIN ×2 (14, 14)	CM, RCC, Meningioma
NL-18	UM & HCC and family history of MMe & HCCs	UM (72) HCC (68)	CM^a^, MMe ×3, NMSC ×5
NL-19	CM and family history of UM	CM (44)	UM, NMSC
NL-20	Familial CM ^b^	CM (45)	CM ×3
NL-21	UM and CM	UM (44) CM (55)	CM ^a^ ×3, RCC
NL-22	Familial CM	CM (47) NMSC ×7 (54, ?) Prostate cancer (55)	CM, NMSC ×2

^a^ Retrospectively found in patients with wild-type (WT) germline *BAP1* status, ^b^
*BAP1* germline variant found in research setting. Abreviations: UM: uveal melanoma, CM: cutaneous melanoma, MMe: malignant mesothelioma, RCC: renal cell cancer, BIN: BAP1-inactivated nevus, NMSC: non-melanoma skin cancer (mostly basal cell carcinomas), CoM: conjunctival melanoma, VS: vestibular schwannoma, HCC: hepatocellular carcinoma. ?: age could not be retraced.

**Table 2 cancers-11-01114-t002:** Germline *BAP1* variants causing a truncated protein.

Family Identifier	Region	Germline Variant	Protein Change	Chromosome Position (GRCh37/hg19)	Pathogenicity, ACMG Classification [36]	Previously Published in	Previously Reported in
NL-1	Exon 1	c.35delC	p.(Pro12fs*)	g.52443860del	Pathogenic	Walpole et al., 2019 [12]	
NL-2	Exon 1	c.35_37+2delinsAGGG	p.(Pro12fs*)	g.52443856_52443860delinsCCCT	Pathogenic		New variant
NL-3	Exon 4	c.178C>T	p.(Arg60*)	g.52442567G>A	Pathogenic	Walpole et al., 2019 [12]	Njauw et al., 2012 [4] Wadt et al., 2015 [36]
NL-4	Exon 4	c.182delA	p.(Lys61fs*)	g.52442563del	Pathogenic		New variant
NL-5	Exon 4	c.188_189delCT	p.(Ser63fs*)	g.52442556_52442557del	Pathogenic		New variant
NL-6	Exon 6	c.376_377delAG	p.(Ser126fs*)	g.52441475_52441476del	Pathogenic		New variant
NL-7	Exon 11	c.1017_1048del ^a,b^	p.(Gly340fs*)	g.52439194_52439225del	Pathogenic	Walpole et al., 2019 [12]	
NL-8	Exon 12	c.1153C>T	p.(Arg385*)	g.52438566G>A	Pathogenic		Njauw et al., 2012 [4] Betti et al., 2016 [37]
NL-9	Exon 13	c.1530delT	p.(Ile511fs*)	g.52437631del	Pathogenic		New variant
NL-10	Exon 13	c.1621delG	p.(Val541fs*)	g.52437540del	Pathogenic		New variant
NL-11	Exon 13	c.1621delG	p.(Val541fs*)	g.52437540del	Pathogenic		New variant
NL-12	Exon 14	c.1768C>T	p.(Gln590*)	g.52437276G>A	Pathogenic	Walpole et al., 2019 [12]	
NL-13	Exon 14	c.1819delA ^b^	p.(Thr607Argfs*)	g.52437225del	Pathogenic		New variant
NL-14	Exon 15	c.1936_1937insTT	p.(Tyr646fs*)	g.52436841_52436842insAA	Pathogenic	Walpole et al., 2019 [12] Potjer et al., 2019 [32]	

^a^ Proband also carried a familial pathogenic variant in *ENG*: c.1116-1117insT (p.Lys373*). ^b^ De novo mutation.

**Table 3 cancers-11-01114-t003:** Germline *BAP1* variants in splice sites and missense variants.

Family Identifier	Region	Germline Variant	Protein Change	Chromosome Position (GRCh37/hg19)	Pathogenicity, ACMG Classification [38]	Previously Published in	Previously Reported in
NL-15	Intron 2	c.67+1G>C	p.?	g.52443729C>G	Likely pathogenic		Repo et al., 2019 [25]
NL-16	Intron 3	c.122+1G>T ^a^	p.?	g.52443569C>A	Likely pathogenic	Potjer et al., 2019 ^a^ [32]	
NL-17	Intron 3	c.122+5G>C	p.?	g.52443573G>C	Likely pathogenic ^b^ [34]		New variant
NL-18	Intron 6	c.437+1G>T	p.?	g.52441414C>A	Likely pathogenic		New variant
NL-19	Intron 8	c.660-2A>G	p.?	g.52440394T>C	Pathogenic	Walpole et al., 2019 [12]	
NL-20	Intron 13	c.1730-1G>A	p.?	g.52437315C>T	Likely pathogenic	Potjer et al., 2019 [32]	
NL-21	Exon 4	c.200A>G	p.(Asp67Gly)	g.52442545T>C	Likely pathogenic [39]	Walpole et al., 2019 [12]	Cabaret et al., 2017 [40]
NL-22	Exon 13	c.1387C>G	p.(Leu463Val)	g.52437774G>C	VUS		New variant

^a^ Proband also carried a (likely) pathogenic variant in *BRIP1*: c.894C>A, p.(Cys298*). ^b^ RNA analysis is described in Appendix A. VUS: Variant of unknown significance.

**Table 4 cancers-11-01114-t004:** Proposed referral guidelines for genetic diagnostics of BAP1-tumor predisposition syndrome (BAP1-TPDS).

Item	Referral Indicated If	
Medical history	≥2 BAP1-TPDS-associated tumors ^a,b^	
Medical history and family history	1 BAP1-TPDS-associated tumor and first- or second-degree relative with ≥1 BAP1-TPDS-associated tumor(s) ^a,b^	
Young age of onset	Uveal melanoma: Cutaneous melanoma: Malignant mesothelioma: Renal cell carcinoma:	age of onset before 40 years age of onset before 18 years [60] age of onset before 50 years [61] age of onset before 46 years [62]

^a^ BAP1-TPDS-associated tumors include: uveal melanoma, cutaneous melanoma, malignant mesothelioma, renal cell carcinoma, meningioma, cholangiocarcinoma, BAP1-inactivated nevus. Non-melanoma skin cancer in case of unusually high frequency in a single individual or at an unusually young age. ^b^ If cutaneous melanoma is the only tumor type, the threshold of initiating germline *BAP1* analysis should be ≥3 cutaneous melanomas in populations with a high incidence of cutaneous melanoma.

**Table 5 cancers-11-01114-t005:** Candidate susceptibility genes per BAP1-tumor-predisposition-syndrome-associated malignancies.

Malignancy	Candidate Susceptibility Genes	References
Uveal melanoma	*MBD4*	[74,75]
Cutaneous melanoma	*CDKN2A*, *CDK4*, *MITF*, *POT1*, *ACD*, *TERF2IP*, *TERT*	[32,72,73]
Malignant mesothelioma	*CDKN2A*, *BRCA2*, *TMEM127*, *VHL*, *WT1*	[30,37]
Renal cell carcinoma	*VHL*, *FLCN*, *SDHB*, *FH*, *MET*, *PTEN*	[9,77,78,79]
Meningioma	*NF2*, *SMARCB1*, *SMARCE1*, *SUFU*	[80,81,82,83,84]
Cholangiocarcinoma	Unknown, possibly *BRCA2*	[85]

**Table 6 cancers-11-01114-t006:** Proposed surveillance recommendations BAP1-tumor-predisposition-syndrome-associated malignancies.

Malignancy	Surveillance as Suggested by Rai et al. [21]		Surveillance as Suggested by Star et al. [86]	
Uveal melanoma	-Dilated eye exams and ophthalmic imaging	-Starting at 11 years annually	-Dilated eye exams, fundus photography, and ocular ultrasound	-Starting at 16 years annually -From 30 years 6 monthly
Cutaneous melanoma	-Full body skin exam and self-skin exam	-Starting at 20 years annually	-Full body skin exam and total body photography	-Starting at 18 years 6 monthly
Malignant mesothelioma	-Physical examinations	-Annually	-Abdominal and respiratory examination	-Starting at age 30
			-Ultrasound exam or MRI -CT or MRI	-Between ages 30–55 biennially -After the age of 55 biennially
Renal cell carcinoma	-Ultrasound -MRI	-Anually -Biennially	Incorporated in mesothelioma surveillance recommendations	

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
