# Peer review of "Families with BAP1-Tumor Predisposition Syndrome in The Netherlands: Path to Identification and a Proposal for Genetic Screening Guidelines"

_cancers, 2019, doi:10.3390/cancers11081114_

Round 1

Reviewer 1 Report

The authors present the BAP1 germline screening in Dutch families with a high risk of BAP1-tumor predisposition syndrome. The BAP1-TPDS is relatively new cancer syndrome; therefore, all new carriers should be published to get a better picture of the syndrome. The results section should present the variants in a more precise way (i.e. all data of the variants are speculated), but otherwise, the manuscript is easy to read.

Major comments:

Pedigrees with the tumours should be added as a supplementary file; figures are easier to read than tables.

Table 3: How it is known that variant c.122+5G>C is likely pathogenic? Five bases from exon is not located in the consensus donor splice site. An explanation should be added to the results section.

Table 3 and results: the canonical +/−1 or 2 splice sites variants should be categorized as pathogenic in a gene where the loss of function (LOF)

is a known mechanism of disease (see the ACMG guidelines, PMID: 25741868)

Table 3: Is the variant c.67+1G>C de novo or do this family have relatives in Finland?

Line 129: Were the paternity tested for these patients? Extramarital pregnancy is possible, and the absence of variants in the parents do not necessarily prove de novo occurrence.

Minor comments:

Line 31: BRCA1 should not be in italics

Line 32: BAP1 should not be in italics in the name of the syndrome trough out the manuscript

Line 356: p-value cannot be negative, "p<0.000"

Reviewer 2 Report

The authors describe the clinical and genetical data of 22 Dutch families with BAP1-TPS. It is well written and quite extensive in most areas. In the discussion the authors propose a recommendation for testing and follow-up which is quite needed. This work is worthwhile publishing and here are a number of minor modifications that are recommended:

1/ In the introduction add leptomeningeal melanoma and paraganglioma to the list of associated tumors (PMID 25900292 and PMID 22889334). These are cited in the 2018 WHO skin cancer textbook in the BAP1-TPS chapter.

2/On lines 104, 223 and778 and tables use the terminology “BAP1-inactivated nevus” instead of inactive.

3/ Regarding the conjunctival MM was BAP1 IHC performed or other molecular testing performed as this is important to rule out an incidental tumor? I think not but should be more clearly outlined as the author put forward this a an important finding in the study but lack clear evidence of relaionshîp (even if statistically suggested).

4/ In the work of Cabaret et al it was suggested as a screening technique to perform BAP1 IHC in other cutaneous nevi and melanomas that were simultaneous or previously removed with a single BIN lesion and this was further recommended by Wadpole et al. When a single BIN is present, this is the most common situation in which a decision for oncogenetic testing is to be made.

Although this study did not perform BAP1 IHC systematically it should be discussed as another screening tool and added to the discussion.

5/ text is a bit lengthy as could be somewhat shortened.
